# Plasma Exosome-Derived microRNAs as Potential Diagnostic and Prognostic Biomarkers in Brazilian Pancreatic Cancer Patients

**DOI:** 10.3390/biom12060769

**Published:** 2022-05-31

**Authors:** Anelis Maria Marin, Sibelle Botogosque Mattar, Rafaela Ferreira Amatuzzi, Roger Chammas, Miyuki Uno, Dalila Luciola Zanette, Mateus Nóbrega Aoki

**Affiliations:** 1Laboratory for Applied Science and Technology in Health, Carlos Chagas Institute, Oswaldo Cruz Foundation (Fiocruz), Curitiba 81310-020, Brazil; anelis.marin@gmail.com (A.M.M.); sibellebm@ibmp.org.br (S.B.M.); dalila.zanette@fiocruz.br (D.L.Z.); 2Laboratory of Expression Regulation, Carlos Chagas Institute, Oswaldo Cruz Foundation (Fiocruz), Curitiba 81310-020, Brazil; rfamatuzzi@gmail.com; 3Center for Translational Research in Oncology (LIM24), Departamento de Radiologia e Oncologia, Instituto Do Câncer Do Estado de São Paulo (ICESP), Hospital das Clínicas da Faculdade de Medicina da Universidade de São Paulo (HCFMUSP), São Paulo 01246-000, Brazil; rchammas@usp.br (R.C.); miyuki.uno@hc.fm.usp.br (M.U.)

**Keywords:** microRNA, exosome, pancreatic cancer, Brazil

## Abstract

Pancreatic cancer represents one of the leading causes of oncological death worldwide. A combination of pancreatic cancer aggressiveness and late diagnosis are key factors leading to a low survival rate and treatment inefficiency, and early diagnosis is pursued as a critical factor for pancreatic cancer. In this context, plasma microRNAs are emerging as promising players due to their non-invasive and practical usage in oncological diagnosis and prognosis. Recent studies have showed some miRNAs associated with pancreatic cancer subtypes, or with stages of the disease. Here we demonstrate plasma exosome-derived microRNA expression in pancreatic cancer patients and healthy individuals from Brazilian patients. Using plasma of 65 pancreatic cancer patients and 78 healthy controls, plasma exosomes were isolated and miRNAs miR-27b, miR-125b-3p, miR-122-5p, miR-21-5p, miR-221-3p, miR-19b, and miR-205-5p were quantified by RT-qPCR. We found that miR-125b-3p, miR-122-5p, and miR-205-5p were statistically overexpressed in the plasma exosomes of pancreatic cancer patients compared to healthy controls. Moreover, miR-205-5p was significantly overexpressed in European descendants, in patients with tumor progression and in those who died from the disease, and diagnostic ability by ROC curve was 0.86. Therefore, we demonstrate that these three microRNAs are potential plasma exosome-derived non-invasive biomarkers for the diagnosis and prognosis of Brazilian pancreatic cancer, demonstrating the importance of different populations and epidemiological bias.

## 1. Introduction

With 495,773 new cases in 2020, pancreatic cancer represents the 12th most common form of cancer worldwide. However, it is the third leading contributor to cancer mortality in the United States, and the seventh in global cancer death [1,2]. The global incidence (5.7 and 4.1 age-standardized incidence rate per 100,000, respectively) and mortality (4.9 and 4.5 age-standardized incidence rate per 100,000, respectively) are higher in men than in women and correlate with increasing age [2]. Furthermore, pancreatic adenocarcinoma is the most common type (85% of cases) arising in exocrine glands, while pancreatic neuroendocrine tumor (PanNET) is less common (less than 5%) and occurs in the endocrine tissue of the pancreas [3].

Pancreatic cancer patients have the lowest survival rate of all major organ cancers, which is closely related to tumor stage [4]. Bengtsson [4] recently analyzed the long-term survival (≥5 years) of pancreatic cancer patients based on the database between 1975 and 2011 of the National Cancer Institute’s Surveillance, Epidemiology, and End Results (SEER). The authors found that 5-year survivors for all stages represented only 0.9% in 1975, increasing to 4.2% in 2011. Recent data from 2021 from the American Cancer Society showed that the 5-year survival rate is 10% for all stages combined [5].

Pancreatic cancer usually undergoes late diagnosis in advanced tumor stages due to a wide range of non-specific symptoms. The more common clinical symptoms are weight loss, jaundice, abdominal pain, anorexia, and dark urine [6]. Image-based tests, abdominal ultrasonography, triphasic pancreatic-protocol (arterial, late, and venous phases) cross-sectional imaging, and magnetic resonance imaging (MRI) are useful tools. Moreover, if a pancreatic mass is identified, subsequent endoscopic ultrasonography and fine-needle aspiration biopsy are indicated for cytological diagnosis [6,7,8,9]. The CA-19-9 antigen can be useful to search for plasma biomarkers associated with diagnosis and prognosis, and recurrence after resection (National Comprehensive Cancer Network 2014). However, since this biomarker is not tumor-specific, CA-19-9 is not a reliable screening tool for individual differences in asymptomatic patients, as it has a sensitivity and specificity of around 75 and 80%, respectively [10,11,12]. Moreover, this biomarker cannot distinguish between cancer and chronic pancreatitis and possibly between other disease states with chronic inflammation [13]. In people lacking the Lewis antigen A (about 10% of the Caucasian population), CA19-9 is not produced by any cells [14].

MicroRNAs (miRNAs) are a class of small non-coding RNAs (17–25 nucleotides) that regulate gene expression post-transcriptionally by repressing translation and/or initiating mRNA degradation [15] and are expressed in several tissues and cell types. The deregulated expression of these molecules impacts health and may favor disease manifestations. MicroRNAs have a critical regulatory function and are found in all physiological processes, including malignant tumors. Therefore, they can be used as diagnostic and prognostic biomarkers during cancer treatment. This function can be applied in other diseases beyond tumors, such as viral infections, nervous system disorders, cardiovascular [16,17] and muscular systems, diabetes, pregnancy complications such as pre-eclampsia [18], and others [19].

Cancer cells and their microenvironment are both responsible for tumor malignancy. In this context, the relationship between exosomes and their microRNA reveals that they are involved in cellular pathways, have a role in cancer as tumor suppressors, and/or act as oncogenes promoting cell proliferation and migration, epithelial–mesenchymal transition (EMT), tumor proliferation, and angiogenesis, metastasis [16]. As with microRNAs, exosome-derived microRNAs affect the tumor environment by influencing the extracellular matrix, and activation and recruitment by the immune system [20]. Nishiwada et al. [21] reveals that is possible to identify an miRNA signature to predicts recurrence following surgery in patients with PDAC (pancreatic ductal adenocarcinoma).

Studies showing miRNAs as potential biomarkers have been recently published [22,23]. Vicentini et al. [22] showed that pancreatic lesions are characterized by a specific exosomal-miRNA signature. In patients with pancreatic cancer, the exosomal miR-451a showed a significant association with tumor size and stage, being highly upregulated in stage II of the disease [24]. Mir196a and miR-1246 were found in elevated levels in pancreatic cancer plasma and analyzing the cancer subtype miR-196a is a better indicator of pancreatic ductal adenocarcinoma (PDAC), whereas plasma exosome miR-1246 is significantly elevated in patients with intraductal papillary mucinous neoplasms (IPMN) [23]. Abue et al. [25] reported that the expression of plasma exosomes miRNA-483-3p was higher in pancreatic cancer patients compared with IPMN patients. Li et al., 2020 [26] identified that miR-5703 in the exosomes acted as a driver of cell proliferation through the binding at CMTM4, downregulating its expression, and promoting the pancreatic cancer cells’ proliferation due to the PAK4 activated PI3K/Akt pathway.

As there are still no plasma exosome-derived miRNAs reported as biomarkers in clinical practice in Brazilian pancreatic cancer, the aim of this study is to correlate miRNAs and clinical data, where this validation report is based on a discovery study from Chinese pancreatic cancer patients [27]. Therefore, this is the first report on plasma exosome-derived microRNAs in pancreatic cancer patients and healthy controls from Brazil. After correlating with clinical data, we demonstrate the role of miRNAs as useful and promising diagnostic tools.

## 2. Materials and Methods

### 2.1. Ethical Statement and Samples

This work was conducted after approval from the Ethics Committee of Fiocruz, Instituto do Câncer do Estado de São Paulo (ICESP) and Hospital do Trabalhador (CAAE 89520218.7.0000.5248, 77979417.8.0000.5248, and 77979417.8.3001.5225). All samplings and experiments were carried out following relevant guidelines, Brazilian regulations, and ethical principles for human research of the Declaration of Helsinki. The project was described to all participants, and a written informed consent and epidemiological questionnaire was obtained from all participants enrolled in the study. A total of 65 pancreatic cancer patients were recruited from 2018 to 2019 with inclusion criteria as disease confirmation by histopathology and/or surgery at the moment of sample collection provided from the Academic Network for Cancer Research Biobank of the University of São Paulo (Biobank-USP), located in the Center for Translational Research in Oncology, São Paulo State Cancer Institute (Centro de Investigação Translacional em Oncologia, Instituto do Câncer do Estado de São Paulo-ICESP), São Paulo, Brazil. The Biobank–USP protocol was approved by the Local Ethics Committee (CEP no. 031/12 and National Ethics Committee (CONEP no. 023/2014). As a control group, 78 non-cancer participants were recruited from Hospital do Trabalhador, Curitiba PR, Brazil, with inclusion criteria of non-personal history of any kind of cancer. Demographic and epidemiological data were collected in both groups, while clinical data were collected for pancreatic cancer patients. For each patient, 4 mL of peripheral blood was collected in EDTA tubes, which were immediately centrifuged at 3000× *g* for 10 min at 4 °C, and had their plasma was separated and frozen at −80 °C.

### 2.2. Exosome Isolation and miRNA Purification

Exosome isolation was performed by the commercial kit miRCURY^®^ Exosome Serum/Plasma Kit (Qiagen, Hilden, Germany) following the manufacturer instructions for an initial volume of 0.6 mL of plasma. The purified exosome samples were stored at −80 °C and a fraction (0.15 mL) was used to total RNA purification using miRNeasy Micro Kit (Qiagen, Germany) following the manufacturer protocol.

### 2.3. Exosome Tracking Analysis

Exosome Tracking Analysis using Nanosight LM-10 (Malvern Panalytical, Malvern, UK) was used to determine the concentration and size of exosome isolated from both blood plasma of pancreatic cancer patients and controls. Exosome samples were diluted in PBS before injection for cell range from 40 to 100 by the reading frame. The videos were set to three runs each of 60 s, detection threshold was defined as 4, and camera level as 12. The data were analyzed using miRNA expression.

The selection of microRNAs was performed following previous studies by Zhou et al. [27], who found some microRNAs upregulated in the plasma of pancreatic cancer. For this reason, a panel of seven microRNA was selected for quantification: miR-27b-3p, miR-125b-3p, miR-122-5p, miR-21-5p, miR-221-3p, miR-19b-3p, and miR-205-5p, as previously reported in the scientific literature [27,28,29]. As reported by Zhou et al. [30], miRNA 103a-3p was selected for housekeeping due to its stable expression in control and tumor samples. The miRNAs extracted from exosomes were reverse transcribed to cDNA using the TaqMan™ Advanced miRNA cDNA Synthesis Kit (Thermo Fisher Scientific, Waltham, MA, USA), which uses a universal reverse transcription to prepare the cDNA template. For miRNA quantification, TaqMan Advanced MicroRNA Assays (Thermo Fisher Scientific, USA) were used as described by the manufacturer, using TaqMan™ Fast Advanced Master Mix (Thermo Fisher Scientific, USA). The reactions were performed on StepOne™ Real-Time PCR System (Thermo Fisher Scientific, USA), in duplicate, using the default cycling conditions recommended by the manufacturer. In the same plate, the control and case (50 to 50%) samples were analyzed to a unique miRNA. For the analysis, miRNA was considered when all replicates had Ct values lower than 38.

### 2.4. Statistical Analyses

NTA statistical analysis was performed with Minitab Statistical Software 17.0, where samples were submitted to one-way variance analysis (ANOVA), and values were compared with Tukey’s test with a 5% probability level. For miRNA expression, a control sample (among those tested) was selected as reference, and 2^−ΔΔCt^ calculation was performed, with the difference between the expression of pancreatic cancer and control samples calculated by *t*-test [31]. One-way ANOVA was used to calculate expression of the same miRNA in pancreatic cancer subgroups. The diagnostic value of miRNAs was analyzed by the receiver operating characteristic curve (ROC) curve, demonstrating area under the curve (AUC), and a 95% confidence interval. A statistically significant value was considered with *p* < 0.05.

## 3. Results

### 3.1. Epidemiological Data

Epidemiological data for ancestry, sex, age, diabetes, alcohol, and tobacco usage for both groups are represented in Table 1. No statistically significant difference was observed between cases and controls for these parameters.

Table 2 shows clinical data for pancreatic cancer patients. Of the total of patients, 75.4% had adenocarcinoma, 7.7% intraductal papillary mucinous neoplasm (IPMN), 4.6% Frantz Tumor, and 12.3% were diagnosed with other types of cancer, such as neuroendocrine tumors and exocrine tumors. Tumor location was predominant in the pancreas head, while 53.7% of patients had lymph node dissection and 47.0% of them were positive for tumor cells. Patients’ progression, tobacco and alcohol usage, and diabetes are also shown in this table, along with treatment data.

### 3.2. Nanoparticle Tracking Analysis

The average size of exosomes of the healthy controls (114.13 ± 13.70 nm) was slightly lower than that of pancreatic cancer patients (117.3 ± 24.68 nm), although with significant variation (F = 3.246; *p* = 0.0351; df = 14) (Figure 1A). Regarding the average concentration of exosomes, a higher and significant difference (t = 6.088; *p* ≤ 0.0001; df = 15.09) was observed between the healthy control groups (8.83 × 10^11^ ± 3.5 × 10^11^ particles/mL) and pancreatic cancer patients (3.70 × 10^12^ ± 1.8 × 10^12^ particles/m) (Figure 1B).

### 3.3. MicroRNA Expression

As mentioned in Material and Methods, miRNAs were considered for analysis when all replicates had Ct values lower than 38. All the reactions were performed with the same reagent lot (master mix and microRNA assays) and no batch effect was observed. Table 3 and Figure 2 show all seven exosome-derived microRNA expressions in pancreatic cancer patients and healthy controls. We can observe that miRNA miR-125b-3p, miR-122-5p, and miR-205-5p have significantly higher expression in pancreatic cancer patients than in healthy individuals.

Additionally, exosome-derived miRNA expression was analyzed according to the type of cancer, ethnicity, tumor progression, and vital status. Table 4 shows that miR-125b-3p, miR-122-5p, and miR-205-5p were significantly overexpressed in pancreatic adenocarcinoma patients than in other types of pancreatic tumors, such as IPMN, Frantz Tumor and PanNET. Interestingly, miR205-5p was significantly overexpressed only in European descendants, patients with tumor progression, and those who died from the disease.

A ROC curve was established to investigate the diagnostic value of these three exosome-derived miRNAs as biomarkers in pancreatic cancer, correlating to healthy controls. Figure 3 shows the ROC curves of miR125b-3p, miR-122-5p, and miR-205-5p. These biomarkers had an AUC of 0.736 (0.644–0.827), 0.726 (0.643–0.809), and 0.829 (0.753–0.904), respectively, all with *p* < 0.0001. When the ROC curves for these microRNAs were generated just for adenocarcinoma patients, the AUC was increased, indicating a better diagnostic ability, with values of 0.782 (0.689–0.875), 0.814 (0.740–0.888), and 0.857 (0.785 to 0.929), for miR125b-3p, 122-5p, and 205-5p, respectively.

An in-silico analysis using DIANA-miRPathv3.0 using Tarbase as a reference to access KEGG pathways related to the three significantly overexpressed miRNA returned significant results for several pathways, including pancreatic cancer (KEGG pathway 05212; *p* = 0.00007). These three miRNAs target 24 genes related to pancreatic cancer pathways, including AKT, TP53, and BRCA2.

## 4. Discussion

Biomarkers for pancreatic cancer diagnosis and prognosis are emerging as useful tools in scientific and clinical contexts, located in exosomes or freely plasma. miRNAs are not only key players and promising minimally-invasive biomarkers, but they also have roles in pancreatic cancer initiation and mechanisms. Therefore, in the present study we investigated exosome-derived miRNAs in pancreatic cancers. Exosomes are secreted by multiple cell types, including tumor cells; they differ in diameter size (40–160 nm), content, and functional effect on recipient cells [32]. Exosome production by cancer cells is usually higher when compared with non-cancer cells, which is probably related to the source of exosomes that influence their heterogeneity [11]. Like other studies on cancer-derived exosomes [11,33], our results show that plasma-derived exosomes from healthy controls and pancreatic cancer patients exhibit equal range size but a contrasting concentration profile.

One of the first reports regarding miRNA in pancreatic cancer tissue was described by Lee et al. [34]. These authors analyzed 28 tumors, 15 adjacent benign tissues, four chronic pancreatitis specimens, six normal pancreas tissues, and nine pancreatic cancer cell lines, where one hundred micro-RNA precursors were aberrantly expressed in pancreatic cancer. Most of the top aberrantly-expressed microRNA displayed increased expression in the tumor, and reverse transcription in situ PCR showed that three of the tops differentially expressed miRNAs (miR-221, -376a, and -301) and were localized in the tumor cells and not in stroma or normal acini or duct cells [34]. Using formalin-fixed paraffin embedded (FFPE) samples, miR-21, miR-155, miR-210, miR-221, and miR-222 were overexpressed in pancreatic adenocarcinoma in comparison to control samples, whereas miR-31, miR-122, miR-145, and miR-146a were underexpressed, and it was shown that the expressions of miR-21 and miR-155 were associated with tumor stage and poor prognosis [35].

In addition, miRNA expression in pancreatic cancer tissues has also been observed in clinical evolution and prognosis analyses. Expression profiles of pre-processed mature miRNA for pancreatic cancer and pancreatic tissues from healthy individuals were accessed from The Cancer Genome Atlas (TCGA) database, based on the IlluminaHiSeq_miRNASeq platform (Illumina Inc., San Diego, CA, USA), and seven downregulated miRNAs in pancreatic cancer (miR-424-5p, miR-139-5p, miR-5586-5p, miR-126-3p, miR-3613-5p, miR-454-3p, and miR-1271-5p) were selected to generate a signature [36]. Based on this seven-miRNA signature, the authors proposed a stratification of pancreatic cancer patients into low- and high-risk groups [36].

It has been shown that miR-216b-5p expression is significantly decreased in pancreatic cancer tissues and cell lines, and its overexpression was positively correlated with pancreatic cancer cell proliferation, induced cell cycle arrest and cell apoptosis in vitro and inhibited tumorigenesis in vivo, acting as a potential tumor suppressor by regulating TPT1 [37]. High expression of miR-301a in human pancreatic cancer patients is related to poor survival, and its overexpression enhances pancreatic cancer cell invasion, angiogenesis, and migration. In contrast, its inhibition suppresses cell invasion of pancreatic cancer and reduces growth and metastasis of orthotopic pancreatic tumors. Furthermore, miR-301a was found to suppress the expression of SOCS5, which leads to JAK/STAT3 activation and is related to poor overall survival of pancreatic cancer patients [38]. Several other reports have demonstrated miRNA playing a biological role in pancreatic cancer pathways, such as those targeting TXNIP-Mediated HIF1α [39], E2F5 [40], TGF-β2/TGF-βRIII [41], HIF1α [42], and PTEN [43].

With scientific and clinical demonstration of microRNA differential expression in pancreatic cancer tissues, a screening for their use as a plasma biomarker for diagnosis and prognosis has emerged as a potential and functional role. In 2014, Schultz et al. [44] provided one of the first reports on this matter, when they observed differences in 38 microRNA were significantly dysregulated in the whole blood between pancreatic cancer patients and healthy participants. The authors suggested there were two microRNA-based diagnostic panels with the potential to distinguish pancreatic cancer patients from healthy control [44]. So far, we have found three plasma exosome-derived microRNAs with higher expression in pancreatic cancer patients in comparison to healthy subjects. Using plasma samples, Zhou et al. [30] identified a six-miRNA signature, including upregulated miR-122-5p and miR-125b-5p in pancreatic cancer patients, and suggested that plasmatic miR-125b-5p might act as an independent biomarker to predict overall survival of pancreatic cancer patients. A recent report used digital PCR (ddPCR) technology in a two-stage process to assess miRNA expression in the plasma from pancreatic cancer patients and healthy controls and compared their diagnostic performance compared to CA19-9 and the associations with patients’ clinical phenotypes and outcomes [45]. A significant overexpression of miR-122-5p, miR-1273g-3p, and miR-6126 in pancreatic cancer patients was found when compared to healthy controls, in line with the trend of the CA19-9 levels, and increased miR-122-5p levels emerged as an independent negative prognostic factor for pancreatic cancer patients [45].

Additional studies on miRNA as minimally-invasive biomarkers for pancreatic cancer used public databases integrating various miRNA serum expression profiles to find helpful candidates for diagnosis and prognosis. Shams et al. [46] selected 27 promising 27 expressed miRNAs and found that miR-1469 and miR-4530 were individually able to distinguish pancreatic cancer with the highest specificity and sensitivity. The authors also created five diagnostic models consisting of different combinations of miRNAs. They found that the combined miR-125a-3p, miR-5100, and miR-642b-3p were the most promising, with 0.95 AUC, 98% sensitivity, and 97% specificity [46]. Savareh et al. [47] investigated circulating miRNAs expression in pancreatic cancer using bioinformatics methods through analyzing four GEO microarray datasets, which were assessed with a machine learning method, using a combinatorial approach consisting of Particle Swarm Optimization (PSO) plus Artificial Neural Network (ANN) and Neighborhood Component Analysis (NCA) iterations. Their final model consisted of miR-663a, miR-1469, miR-92a-2-5p, miR-125b-1-3p, and miR-532-5p, which showed a great diagnostic ability (93% accuracy, 93% sensitivity, and 92% specificity), and Kaplan–Meier survival assessments revealed that miR-1469, miR-663a, and miR-532-5p was significantly associated with the prognosis of pancreatic cancer patients [47].

The search for miRNA that can be used as minimally-invasive biomarkers in the plasma of pancreatic cancer presents variable results in the scientific literature due to variation in population genetics, as well as clinical properties and treatment of pancreatic cancer. In the present study, we found plasma exosome-derived miR-125b-3p, miR-122-5p, and miR-205-5p were significantly overexpressed in pancreatic adenocarcinoma in comparison to healthy controls, and that miR27b, miR-21-5p, miR-221-3p, and miR-19b had no difference. In this regard, miR-21, one of the top players in the oncological field [48] showed no difference of expression in pancreatic cancer plasma exosomes. A serum six-microRNA signature generated by Zou et al. [49] found no evidence of miR-21 as a participant, in line with the findings of Ye et al. [50] and Ma et al. [51]. In contrast, a recent study demonstrated that the expression of exosome-derived miR-21 and miR-210 was significantly higher in patients with pancreatic cancer than in healthy controls [52]. Similarly, Pu et al. [53] found significantly greater expression levels of plasma exosome-derived miR-21 and miR-10b in the group of pancreatic cancer patients than the control group. In addition, the authors observed that miR-21 could distinguish patients at the early stage of the disease from healthy controls and from advanced-stage patients [53]. With the same research bias context, it has been reported that the levels of plasma exosome-derived miR-19b, normalized using miR-1228, were significantly lower in pancreatic cancer patients than in healthy volunteers [54]. Its diagnostic value was superior to that of serum CA19-9 [54], but reports have also shown that miR-19b is significantly overexpressed in the serum from pancreatic cancer patients [49]. In the present report, the expression of miR-19b showed no difference.

Here, we investigated for the first time, plasma exosome-derived microRNAs in Brazilian pancreatic cancer patients and showed that miR-125b-3p, miR-122-5p, and miR-205-5p had significantly higher expression. Moreover, we found that expression of miR205-5p also was correlated with tumor progression and survival status in pancreatic cancer patients. This report highlights the importance and promising role of plasma, exosome-derived microRNAs in pancreatic cancer diagnostic and prognosis, especially regarding different populations. Given the epidemiological bias for such analyses, we emphasize the need to look for personalized panels of exosome-derived microRNAs, according to the study population.

In this study, we considered some limitations that should be discussed. Regarding the use of a Cycle Threshold lower than 38 for miRNA analysis, we first highlight that most of the reports about exosomal-derived microRNA expression lack this information [55,56], and furthermore, some relevant scientific works use a Ct value lower than 37 [27,57]. In this context, we performed the analyses with Ct < 37 and the results remained the same than with Ct < 38, showing our result is reliable and consistent with similar reports found in the literature. Moreover, we used TaqMan Advanced MicroRNA Assays and not intercalating dye, such as SYBR Green, where high Ct for this second choice could indicate a false-positive or fluorescence background. Another concern is about high variation on exosome-derived microRNA expression between intragroup samples, generating a high standard deviation. This is partially explained by subject differences, such as disease stage and treatment, where when we subdivided the pancreatic cancer patients in specific groups, we observed a standard deviation reduction (Table 4), suggesting that exosome-derived microRNAs are highlighted to be considered with stratified and individualized disease contexts. The last limitation is related to miR205-5p expression, where its expression was significantly different between groups; however, its presence was absent in many pancreatic cancer patients and control subjects, and in fact, this absence represents a negative point regarding the use of miR205-5p as a plasmatic biomarker for pancreatic cancer. However, we believe that exosome-derived microRNAs as plasmatic biomarkers should be based on a panel, containing several microRNAs that should be analyzed together and according to the patient disease stratification.

## Figures and Tables

**Figure 1 biomolecules-12-00769-f001:**
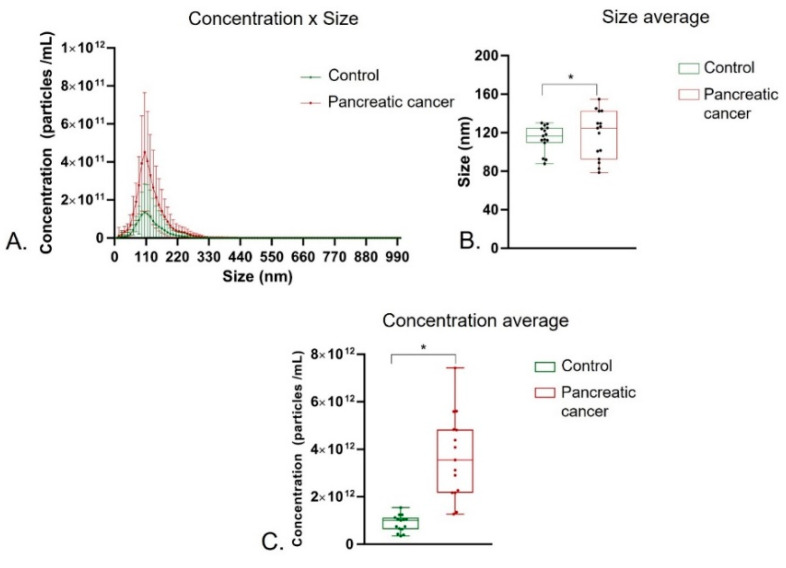
Nano tracking analysis of plasma exosomes isolated from healthy individuals and pancreatic cancer patients. (**A**) Concentration average of exosome population observed between control group and pancreatic cancer patients. (**B**) Average size of exosome population observed between control group and pancreatic cancer patients. (**C**) Comparation of population concentration profile exosome plasma, from control group and pancreatic cancer patients (**A**,**B**). The results were analyzed by *t*-test, comparing the plasmas of patients with the control. (* *p* ≤ 0.05). EV: extracellular vesicle.

**Figure 2 biomolecules-12-00769-f002:**
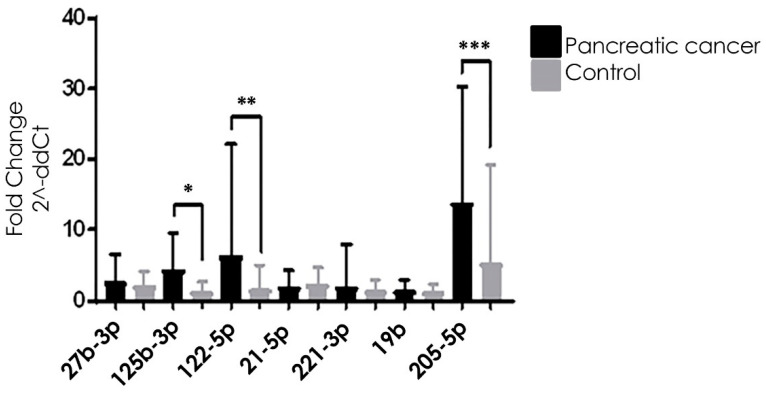
Relative microRNA expression comparison between pancreatic cancer and healthy control groups. miR-125b-3p*, miR-122-5p**, and miR-205-5p*** were expressed significantly higher in pancreatic cancer groups than healthy control groups (* *p* <0.0001; ** *p* = 0.0121; *** *p* = 0.0091).

**Figure 3 biomolecules-12-00769-f003:**
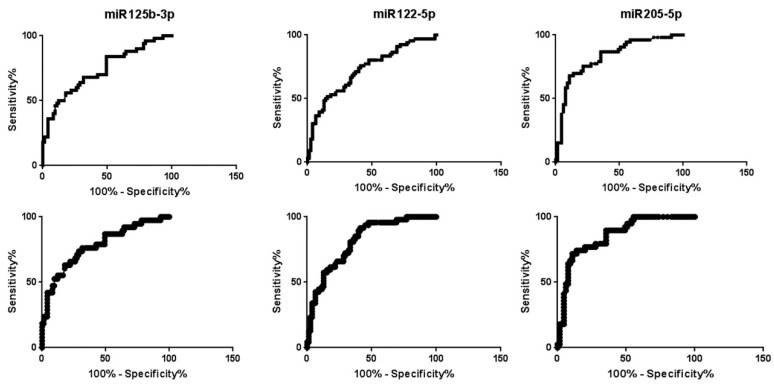
The upper graphics represent ROC curves of miR125b-3p, 122-5p, and 205-5p for overall pancreatic cancer and healthy controls, with AUC values of 0.736, 0.726, and 0.829, respectively. The lower graphics represent ROC curves of miR125b-3p, 122-5p, and 205-5p only for adenocarcinoma pancreatic cancer and healthy controls with AUC values of 0.782, 0.814, and 0.857.

**Table 1 biomolecules-12-00769-t001:** Epidemiological data for pancreatic cancer and healthy control for ancestry, gender, age, tobacco, and alcohol usage and diabetes.

	Ancestry		Sex		Age	Tobacco		Alcohol		Diabetes	
	European	African	Male	Female	Mean ± SD	Yes	No	Yes	No	Yes	No
Pancreatic cancer	66%	34%	40%	60%	60.72 ± 13.3	45%	55%	26%	74%	34%	66%
Healthy controls	71%	29%	27%	73%	63.74 ± 10.2	35%	65%	28%	72%	10%	90%

SD: standard deviation.

**Table 2 biomolecules-12-00769-t002:** Clinical and epidemiological data for pancreatic cancer patients, according to cancer type, tumor location, tumor progression, tobacco and alcohol usage, diabetes, and treatment.

Pancreatic Cancer Type	(%)	
Adenocarcinoma	75.4%	
IPMN	7.7%	
Frantz Tumor	4.6%	
Other	12.3%	
**Tumor location**	**(%)**	
Head	67.37%	
Body	12.63%	
Tail	6.32%	
Other	13.68%	
Dissected lymphonodes	53.68%	
Positive lymphonodes	47.06%	
**Treatment**	**(%)**	
Folfirinox	29.23%	
Gemcitabine	3.08%	
Modified FLOX	10.77%	
Gemzar	3.08%	
Other	1.54%	
Follow	21.54%	
Surgery	7.69%	
Surgery + chemoterapy	23.08%	
	No (%)	Yes (%)
Progression	66.32%	33.68%
Smoking	55.79%	44.21%
Ethanol	72.63%	27.37%
Diabetes	68.42%	31.58%

IPMN: intraductal papillary mucinous neoplasm.

**Table 3 biomolecules-12-00769-t003:** Statistical evaluation of miRNA expression in pancreatic cancer patients (PC) compared to the healthy control group (Ctl). Significantly higher expression was observed in miR125b-3p, miR122-5p, and miR205-5p in pancreatic cancer patients than healthy controls. The values are calculated by 2^−^^ΔΔ^^Ct^ (StdDev: Standard Deviation).

	MiR-27b-3p		MiR-125b-3p		MiR-122-5p		MiR-21-5p		MiR-221-3p		MiR-19b		MiR-205-5p	
	PC	Ctl	PC	Ctl	PC	Ctl	PC	Ctl	PC	Ctl	PC	Ctl	PC	Ctl
N	52	77	**48**	**69**	**57**	**74**	65	78	61	78	61	77	**44**	**61**
Mean	2.669	2.161	**4.459**	**1.361**	**6.402**	**1.629**	2.009	2.261	1.996	1.52	1.433	1.347	**13.8**	**5.346**
StdDev	3.859	2.015	**5.149**	**1.315**	**15.67**	**3.421**	2.363	2.454	6.054	1.448	1.438	0.9844	**16.65**	**13.87**
*p*-value	0.3311		**<0.0001**		**0.0121**		0.5352		0.5033		0.6757		**0.0091**	

**Table 4 biomolecules-12-00769-t004:** Exosome-derived miR125b-3p, miR-122-5p and miR-205-5p with significantly differential expression in pancreatic cancer patients and healthy controls, divided according to cancer type, ancestry, tumor progression, and survival status.

	miR-125b-3p			miR-122-5p			miR-205								
	Pancreatic Cancer Type			Pancreatic Cancer Type			Pancreatic Cancer Type			Ancestry		Tumor Progression		Survival Status	
	Control	Adeno	Other	Control	Adeno	Other	Control	Adeno	Other	African	European	No	Yes	Alive	Dead
Samples	73	**38**	12	78	**47**	18	65	**39**	14	17	**36**	36	**17**	35	**18**
Mean	1.41	**5.32**	2.50	1.55	**7.78**	2.13	5.09	**21.04**	9.96	12.32	**20.85**	11.31	**32.53**	10.95	**32.05**
Standard Error of Mean	0.18	**0.95**	0.90	0.38	**2.51**	0.71	1.67	**5.95**	2.80	2.62	**6.45**	1.66	**13.08**	1.72	**12.31**
Lower 95% CI	1.06	**3.40**	0.52	0.79	**2.74**	0.63	1.75	**8.99**	3.90	6.77	**7.75**	7.94	**4.80**	7.46	**6.08**
Upper 95% CI	1.77	**7.24**	4.47	2.30	**12.82**	3.62	8.43	**33.09**	16.01	17.86	**33.95**	14.68	**60.25**	14.43	**58.02**
*p*-value		**0.0001**	0.9616		**0.0451**	0.9998		**0.0301**	0.9895	0.8992	**0.0404**	0.8378	**0.0024**	0.9412	**0.0038**

## Data Availability

The data presented in this study are available on request from the corresponding author.

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
