# Peer review of "Plasma Exosome-Derived microRNAs as Potential Diagnostic and Prognostic Biomarkers in Brazilian Pancreatic Cancer Patients"

_biomolecules, 2022, doi:10.3390/biom12060769_

Round 1
Reviewer 1 Report
In this manuscript authors showed plasma exosome-derived microRNA expression in pancreatic cancer patients and healthy individuals from Brazilian patients. Using plasma of patients with pancreatic cancer and healthy controls, authors isolated plasma exosomes and quantified miR-27b, miR-125b-3p, miR-122-5p, miR-21-5p, miR-221-3p, miR-19b and miR-205-5p. They found that miR-125b-3p, miR-122-5p and miR-205-5p were increased in plasma exosomes of patients with pancreatic cancer. In particular, miR-205-5p was significantly overexpression in patients with tumor progression, and those who died from the disease showing a good diagnostic ability with an AUC=0.86.
The manuscript is clear and generally well written. Only few points need improvements:
- line 70: authors should point out that miRNA are also important early markers of pregnancy complications. In particular, miR-125b (also studied by the authors) showed an important role in predicting Preeclampsia (PMID: 32726711)
- Figure 2 must be improved and "miR-" to each miRNA must be added
- Figure 4: AUC values should be added under each ROC curve and the reference line should be reported in each graph
Author Response
File attached.

Reviewer 2 Report
In this study, the authors tried to validate a set of plasma exosomal miRNA biomarkers for pancreatic cancer detection with a Brazilian cohort. The miRNA biomarker candidates were adapted from a published work. This reduced the novelty of this work. Based on their analysis, several miRNA candidates showed some potential capability to discriminate between pancreatic cancer patients and healthy controls. However, the are some big issues with the study design and data analysis, which can lead to biased conclusions. And the manuscript was not prepared very well.
The first issue is the cohort used in this study. The inclusion and exclusion criteria for cancer patients and the control subjects are unclear. It is critical to set the criteria before the study was carried out. This will determine the usage of your potential biomarkers in the real clinical settings. It is one of the reasons that there is no exosomal miRNA biomarker has been employed in clinical practice yet. However, a lot of miRNA biomarkers were published for cancer detection or screening.
Another big concern is how to choose the biomarker candidates for validation. As I mentioned previously, this is not a biomarker discovery study. The authors should summarize the recent advances in the field of plasma exosomal miRNA biomarkers for detecting PC in the introduction and abstract. And clarify the reason why only choose the candidates from the paper published in 2018.
The biggest issue of this manuscript is the data analysis. A total 65 PC and 78 control subjects were included in this study, but there are big differences in the N numbers of each tested target in table 3. Surprisingly, too many samples (about 1/3 of subjects in the miR-205 PC group) cannot be included in the analysis, even with a very high Ct threshold of 38. More importantly, all three candidates with significance showed big StdDev in the cancer group or control group. This kind of biomarker candidate is not practical and should be excluded initially.
Some detailed information about the miRNA detection method and data analysis was missing or not considered when the study was performed. For example, did the authors consider the QC of extracted exosomal RNA? Why did they set the Ct threshold so high? Why only 15 paired samples were analyzed in Figure 1 B and C? What are the raw Cts of each target and the control miR 103a-3p in all tested subjects? All the information is important to evaluate the performance of the assay and the data analysis.
All the issues mentioned above raised serious concerns about the reliability of the data analysis and the conclusions.
Author Response
File attached.

Reviewer 3 Report
An original work by Marin et al. entitled ‘’ Plasma exosome-derived microRNAs as potential diagnostic and 2 prognostic biomarkers in Brazilian pancreatic cancer patients ‘’ reveal that expression of specific miRNAs were up-regulated in exosomes of plasma samples of patients. Interestingly, authors found that expression of miR-205-5p was significantly increased in European descendants. These result is important and useful for readers. Authors reported the interesting outcomes, however, they should be caution and rephrase conclusion both in the abstract and conclusion section, ‘’Brazilian pancreatic cancer demonstrating the importance of different populations and epidemiological bias …..I provided some comments on the manuscript. Concerns should be addressed by authors.
1. English grammar and typo errors must to be corrected.
2. What is reason behind selecting miRNAs? Whether authors referenced previous studies or selected onco-miRs specific to pancreatic tumor tissue samples?
3. Why the expression of exosomal miR-205-5p increased in European descendants?
4. Authors provide exosomes characterization methods and results. Exosomes size and markers or images are needed.
5. Provide reference for ‘’ Recent data from 2021 from the American Cancer Society 45 showed that the 5-year survival rate is 10% for all stages combined (American Cancer 46 Society, 2021)’’ add link or DOI.
6. Author should cite references (PMID: 31140622 ; PMID: 33451365 ; PMID: 30051492).
7. Reduce effectively discussion section.
Author Response
File attached

Round 2
Reviewer 2 Report
The significance and novelty of a biomarker study depend on the sophisticated study design and solid data analysis. First of all, it is critical to choose the right reference gene. The authors chose miRNA 103a-3p as the one in the published discovery study. Based on the raw data provided in their responses, the expression level of miR103a-3p showed significant differences (P<0.001) between PDAC group and the control group. Thus, it is not an appropriate housekeeping small RNA for normalization with 2-ΔΔCt method.
The authors have some arguments about the big variations were resulted from the disease stages and treatment. But this cannot explain the reason for similar situation in the control group. This is more like a technique issue. Using a qPCR assay with LOD as low as 3 copies/reaction, you can set the threshold Ct as high as 37 to get a reliable verdict of positive or negative. To get a good linear range in a biomarker investigation, the Ct value is better less than 34. This is the technical reason for the big variation of the replicates that can be identified in your raw Cts.
miR205-5p is non-detectable in 20/65 in the PDAC group and 17 out of 78 healthy samples. How can you classify the non-detected sample If this candidate is used in a clinical setting? On the other hand, non-detectable samples could also mean the copy number of the targets is very low. Excluding such a big proportion of samples as outliers will generate significant bias. Generally, a qualified biomarker candidate for further validation should be detectable at least in 95% of the tested samples.
The big challenge for exosomal RNA as targets for amplification is the very low amount of RNA material. It is very difficult to quantify the starting material used in the RT-qPCR assay. That’s one of the reasons for the high Ct values. One solution is to include a pre-amplification step in the workflow.
As I mentioned, this is a validation study, the original biomarker discovery paper with the Chinese cohort should be mentioned in the introduction. But not only cite it in the method section.
Author Response
File attached
